# Delayed delivery of antibiotics by ultrasound-mediated rupture of polylactic acid pockets: *In vitro* and *in vivo* studies

Selin Isguven Billmyer[1,2], Priscilla Machado[1], Ryan E. Tomlinson[2], Lauren J. Delaney[1], Ji-Bin Liu[1], Alexander H. Harris[2,3], Eric McLaughlin[2], Noreen J. Hickok[2], Flemming Forsberg[1]*

1 Department of Radiology, Thomas Jefferson University, Philadelphia, Pennsylvania, United States of America, 2 Department of Orthopedics, Thomas Jefferson University, Philadelphia, Pennsylvania, United States of America, 3 Cornell University, Ithaca, New York, United States of America

* flemming.forsberg@jefferson.edu

## Abstract

Surgical site infections are a devastating complication of instrumented orthopaedic surgery, particularly in the spine. Bacterial biofilms, once formed on the implant surfaces, exhibit antibiotic tolerance and immune escape, which lead to treatment challenges. Up to 9% of instrumented spine surgeries result in infection, despite the use of systemic antibiotics and local, powdered antibiotics for prophylaxis. The bacteria that survive the initial prophylaxis may be susceptible to a second, high dose of antibiotic prophylaxis prior to establishing at the site. Hence, we designed a local drug delivery system consisting of polylactic acid (PLA) film pockets that can be noninvasively triggered (*i.e.*, ruptured) by the external application of ultrasound (US) following a delay of up to 6 days. We found that thin PLA films (24 ± 4.5 μm) with embedded vancomycin (VAN) powder assembled in a conical pocket shape best allowed for stability and rupturability of this US-mediated delivery system *in vitro* (92% US-triggered rupture of PLA-VAN pockets vs 31% of neat PLA pockets, $p < 0.0001$). VAN-embedded PLA films exhibited decreased strength and toughness (3.47 J/mm³ vs 0.68 J/mm³, $p < 0.0001$) and surface VAN was rapidly released upon submersion, adding an additional layer of protection against bacterial colonization of the device. Finally, a pilot *in vivo* study in five rabbits demonstrated the feasibility of the design, but the stability of neat PLA and PLA-VAN pockets were varied (9/15 of all pockets were intact by Day 3). All of the 5 intact pockets (3 neat PLA and 2 PLA-VAN) that were allocated for insonation ruptured following US (100%). Overall, this design has the potential for use in targeting orthopaedic infections as well as for US-triggered bolus drug release for prophylaxis in high risk cases.

**Data availability statement:** Data generated and/or analyzed during this study are available as a supplemental file with this submission.

**Funding:** NIH R01 AR069119 The Mullin Fund for Spinal Research at Thomas Jefferson University The funders had no role in study design, data collection and analysis, decision to publish, or preparation of the manuscript.

**Competing interests:** "I have read the journal's policy and the authors of this manuscript have the following competing interests: Selin Isguven Billmyer, Noreen Hickok, Flemming Forsberg report financial support was provided by National Institutes of Health. Flemming Forsberg reports a relationship with GE HealthCare that includes: consulting or advisory. Flemming Forsberg reports a relationship with Lantheus Medical Imaging Inc that includes: consulting or advisory. Selin Isguven Billmyer, Noreen Hickok, Flemming Forsberg have patent PCT Application No. PCT/US2023/077854 pending to Thomas Jefferson University. The other authors declare that they have no known competing financial interests or personal relationships that could have appeared to influence the work reported in this paper. This does not alter our adherence to PLOS One policies on sharing data and materials.

## Introduction

Implant-associated infections are dire complications and make up 65% of all bacterial infections in healthcare [1]. In the spine, infections occur in up to 9% of spinal surgical cases [2] and when recalcitrant, can cause death. These infections are associated with biofilms, which, once formed, are difficult to eradicate due to acquisition of antibiotic tolerance and evasion of immune surveillance [3]. To prevent initial biofilm formation and subsequent progression to infection, prophylaxis with systemic antibiotics [2,4–6] is used, with additional placement of local, prophylactic intraoperative antibiotics (powdered antibiotic) [7–9] in many orthopaedic surgeries (in the United States).

Countering this aggressive prophylaxis is the ability of the bacteria to respond to the stressful operative/antibiotic-rich environment by acquiring a "persister" phenotype, *i.e.*, a metabolically suppressed population that are able to survive even high levels of antibiotics. It has been shown that treatment of adherent bacteria require significantly higher (x1000) levels of antibiotics compared to non-adherent bacteria, and, to a lesser extent, high levels are also required for prevention of biofilm formation [1,10,11]. Based on *in vitro* experiments, removal of stressors such as antibiotics allows these bacterial persisters to revert to their original antibiotic-sensitive phenotype, suggesting that delayed antibiotic delivery may have utility [12,13].

Local delivery systems for joint surgeries begin drug elution upon placement into the surgical site [14,15] so that the highest levels of antibiotics elute during the time that systemic and other local antibiotics are at their highest. Hence, we designed a local delivery system to initiate antibiotic release at post-surgical times > 48 hours so as to target bacteria that had survived the initial prophylaxis [16] and provide the pause in challenge to allow for reversion from the persister phenotype. By producing drug release after a brief interlude, the hypothesis is that the second treatment will decrease bacterial presence sufficiently to ultimately allow clearance by the immune system. Finally, based on our previously-tested drug reservoir [11], a biodegradable antibiotic delivery system that can be noninvasively triggered by application of ultrasound (US) was designed [17,18]. In this manuscript, the design, *in vitro* characteristics, and *in vivo* performance of this system are described.

## Materials and methods

### Delivery system design

In spinal surgeries, vancomycin (VAN) powder is thought to be depleted rapidly and dissipated by the time of removal of the wound drains, which usually occurs within the first three days [19,20]. Hence, the drug delivery system was designed to be triggered by US within a 3–5 day period after surgery (Fig 1).

The design criteria for the drug reservoir system were that it was (1) comprised of biocompatible and biodegradable materials, (2) responsive to US-triggered drug release, (3) watertight/intact until ruptured at 3–5 d post-implantation, and (4) had sufficient volume for delivering high levels of antibiotic.

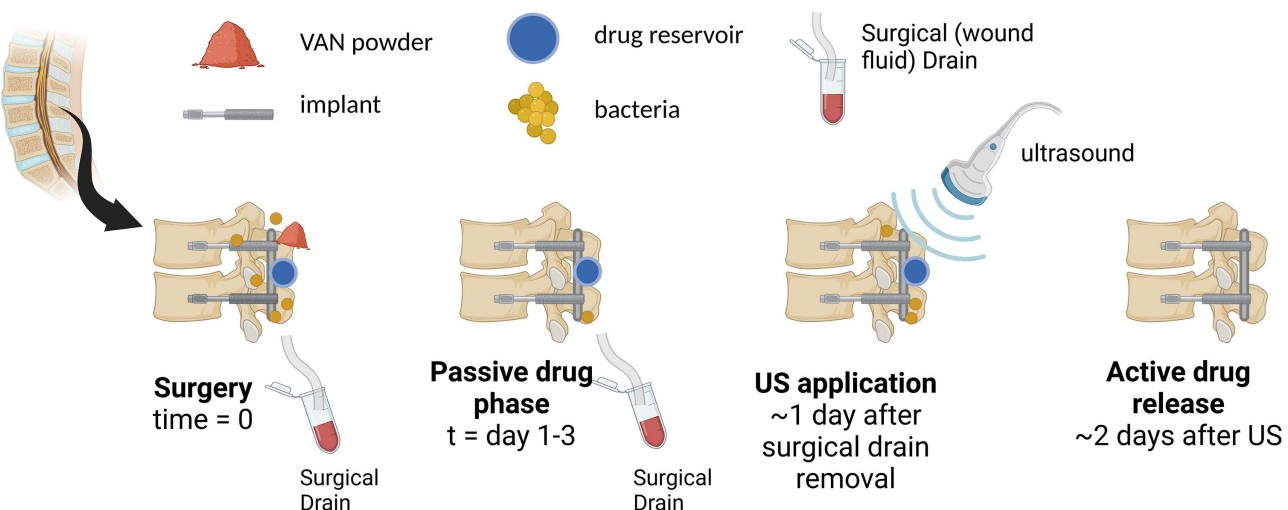

**Fig 1. The timeline of proposed local drug delivery method.** At the time of implant surgery, drug reservoir and VAN powder are placed for local drug administration. The VAN powder dissolves and dissipates rapidly, with the wound fluid removal via drains. Subsequently, US is applied to initiate the delivery of antibiotics.

## Polylactic acid (PLA) film processing

Polylactic acid (PLA) was obtained from several sources, all of which produced similar results ((1) Resomer Select 100 DL 7E (Evonik Industries, Essen, Germany), (2) Standard, natural 1.75 mm PLA filament (3DomFuel, Fargo, ND, USA), (3) Ingeo Biopolymer 2003D, and (4) Ingeo Biopolymer 4060D (NatureWorks, Plymouth, MN, USA)). PLA was dissolved in chloroform (Fisher Scientific, Hampton, NH, USA) with slow mixing, followed by casting onto a 346 cm$^2$ non-stick ceramic surface. Thin films contained 0.5–1.5 g PLA in 30–40 mL chloroform, and thick films, 3.0–5.0 g PLA in 70–80 mL chloroform [11]. Dry PLA film thickness was measured with a digital micrometer (Holite, Inc.). The thinnest viable film was 0.5 g PLA (in 40 mL chloroform), thickness of 24 ± 4.5 µm ($n = 7$); the thick film was 3 g PLA (in 70 mL chloroform) with thickness of 170 ± 25.5 µm ($n = 12$).

## PLA film pocket assembly

PLA film pockets were comprised of a thick PLA film base sealed with a thin (rupturable) PLA film (Fig 2). Dried, (overnight, ~25°C) thick films were cut to 6 cm, folded into a cone for ease of assembly, and the shape stabilized by gluing with cyanoacrylate (The Gorilla Glue Company, Cincinnati, OH, USA) brushed along two inner fold surfaces. The thin, rupturable film lay across the top of the cone. This thin film was stabilized with a "donut" of thick film (outer diameter 4 cm, thickness 1.00–1.25 cm, inner diameter 1.50–2.00 cm) that had been adhered to a partially-dry, semi-sticky thin film. The cone was sealed by gluing the donut layer to the thick film, conical layer.

The vibrantly colored methylene blue (MeB) was used as a drug surrogate [21]. Using a syringe, the pocket assembly was injected with 2.0–4.5 mL of MeB solution up to 80 mg or 4.0 mL, dissolved in deionized (DI) water or phosphate-buffered saline (PBS), through the thick cone and sealed with glued PLA film.

To increase the likelihood of US rupturing the PLA pockets, cavitational nuclei (0.5–4.0 mL) were used to amplify the acoustic energy of US. These nuclei undergo expansion and contraction cycles under acoustic energy (stable cavitation) and ultimately disintegrate irreversibly (inertial cavitation) [22]. The resulting shear waves, microstreaming, microjetting, and shockwaves can cause temporary pores in cell membranes and vessels (*i.e.*, sonoporation) [18]. In these experiments, Sonazoid microbubbles (GE HealthCare, Oslo, Norway) or acoustic nanodroplets derived from Definity

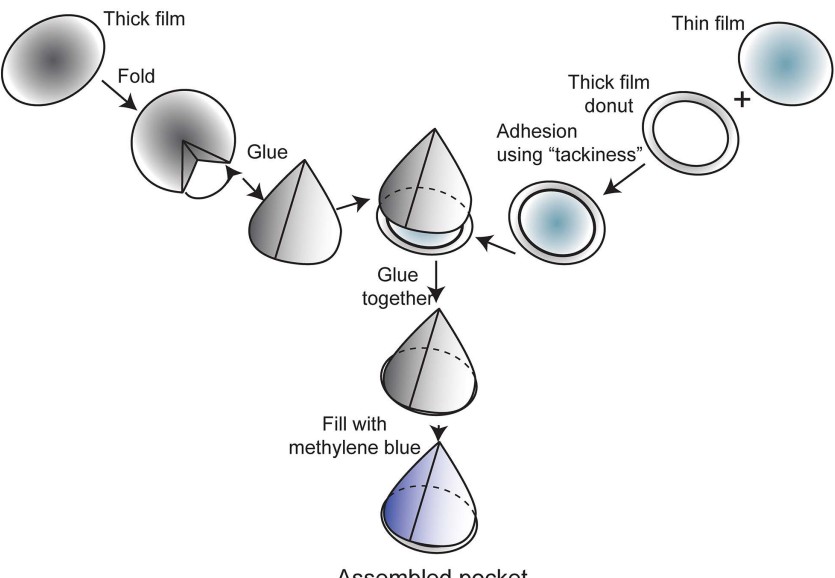

**Fig 2. PLA film pocket assembly; PLA film pocket pieces, with a thick base and a thin rupturable layer with donut.** Circle sizes are not to scale.

microbubbles (Lantheus Medical Imaging, N. Billerica, MA, USA) [23,24], were co-injected into the reservoir. Both bubbles were reconstituted according to instructions from the manufacturer. To form droplets, 1 mL of Definity microbubbles were added to 9 mL 0.9% normal saline and then submerged in the syringe in a cooling bath (−35°C). Because Definity-derived acoustic nanodroplets are smaller in size than Sonazoid microbubbles and the local effects resulting from their cavitation cause a bigger change in volume and perhaps amplification of the cavitation effects such as shock waves, we hypothesized that pockets with Definity droplets would show greater release.

## Incorporation of VAN into the thin PLA film

To facilitate pocket rupture by introduction of local defects, VAN salt, which is insoluble in chloroform, was added to the thin film solution. Specifically, 10–50 mg VAN, which had been ground to a fine powder with a mortar and pestle, was incorporated into the film by (1) dispersing VAN into the PLA-chloroform solution before casting or (2) manually sprinkling VAN onto the drying film in the pan to minimize VAN clumping. These methods were grouped together as "embedded VAN" for analysis.

## Mechanical analysis of PLA films

Neat vs VAN-embedded PLA film groups underwent mechanical analysis. 0.50 g PLA (3DomFuel) films with and without 20–50 mg powdered VAN were cast, dried for 4 weeks, and cut into 0.5 x 2.5 cm test strips. Each sample was clamped securely with thin film grips (FC-20, Imada, Northbrook, IL, USA). The gauge length of approximately 0.5 cm was determined before applying a monotonic displacement ramp of 0.1 mm/s until failure using an ElectroForce 3200 Series III instrument (TA Instruments, New Castle, DE, USA) equipped with a 225 N load cell. Force and displacement were acquired at 25 Hz across 10 neat PLA and 11 PLA-VAN samples using WinTest version 7.2 (Waters Corporation, Milford, MA, USA). Stress was calculated by dividing load over area (calculated as thickness x width (5 mm)), and strain was calculated by dividing the displacement over the gauge length (5 mm). Toughness was computed as the area under the curve (AUC) of the stress-strain plot.

## Atomic Force Microscopy (AFM) analysis of PLA films

To further evaluate film characteristics, we performed atomic force microscopy (AFM) of neat PLA and VAN-embedded PLA films to observe surface topography and visualize VAN inclusions. Film samples were prepared by excising 10 x 3 cm sections from a 19 cm diameter sheet and imaged using an MFP-3D Infinity BIO AFM integrated with a Nikon inverted fluorescence microscope (Asylum Research, Santa Barbara, CA, USA). Images were acquired in AC mode using a triangular gold coated silicon nitride pyramid tip probe (15 nm tip radius, force constant 0.32 N/m, PNP-TR-Au cantilever 1, Asylum Research, Santa Barbara, CA, USA) over a 20 x 20 μm area with 256 lines/256 points at a scan rate of 0.95 Hz. All images obtained were processed using AR software (Igor Pro, Wavemetrics, Portland, OR, USA) integrated with the MFP-3D-BIO AFM. Root mean square (RMS) roughness was quantified using four 10 x 10 μm area regions within each scan from the Z-sensor (height) channel after flattening.

## VAN release from VAN-embedded PLA film

To determine VAN elution in films with embedded VAN, two trials were conducted. In both trials, elution occurred in a sealed container of 50 mL PBS in a 95 rpm, 37°C shaking incubator. In the first trial, 0.5 g PLA film with 50 mg of VAN embedded (high concentration) were cut into pieces corresponding to 2.5 mg of VAN on each piece ($n = 3$). Eluted VAN was measured out to 15 days, with time points every half hour for the first 2 hours, and then sampling at Day 1 and 15. In the second trial, ~20 mg of embedded VAN (low concentration) was present, based on surface area, where sample "d" was approximately 2% and samples "e" and "f" were approximately 1% of the total surface area ($n = 3$). Samples were collected every half hour for 2 h, followed by sampling at 1, 3, 4, and 27 days.

At each time point, a 1 mL sample was collected and replaced with fresh PBS and stored at 4°C. Upon completion, 200 μL of the samples were placed in UV Flat Bottom 96-well Microtiter plates (Thermo, Weaverville, NC, USA) and the VAN concentration was measured spectrophotometrically ($λ = 281$ nm [25] (Infinite M1000, Tecan, Männedorf, Switzerland)). The concentration of VAN in the samples was calculated based on a standard curve of 0–400 μg/mL VAN, run concurrently for each trial.

## US-triggered drug release experiments from PLA film pockets *in vitro*

US-triggered drug release was compared according to rupturable film thickness, presence of VAN powder, type of cavitation agent (as described previously), and type of US ($n = 48$, except for analysis of cavitation agent variable, where $n = 45$ due to 3 pockets not having agents). Pockets (rupturable film (0.5–1.83 g PLA); sealing conical film (3–5 g PLA); reservoir (2–4.5 mL)) were loaded with MeB solution and cavitational agents and tested for leaks; if MeB release was apparent, pockets were discarded. Pockets were submerged in a DI water or PBS bath (400 mL or 2 L) and insonated with either clinical US or high-intensity focused US (HIFU). For clinical US, an S50 scanner (SonoScape, Shenzhen, China) with a curvilinear C1-6 probe was used for 20 min of Power Doppler imaging (2.2 MHz, highest line density, 100% power or 703.6 kPa peak to peak pressure) followed by 10 min of flash replenishment imaging (3.0 MHz harmonic imaging at 100% power or 1.08 MPa peak to peak pressure every 4 seconds). These parameters were measured with a calibrated 0.5 mm needle hydrophone (Precision Acoustics, Dorchester, UK) or selected based on prior experiments [11]. HIFU involved 20 min of insonation (2.0 MHz at 4 V equivalent to 5.5 MPa peak negative pressure with 50% duty cycle) using an SU-101 probe (Sonic Concepts, Bothell, WA, USA) run by an 8116A pulse generator (Hewlett Packard, Palo Alto, CA. USA) with 50 dB amplification.

Initially, these experiments determined US-triggered drug release as a binary yes-or-no outcome by visualization of MeB release immediately upon insonation. Subsequently, we started to incubate pockets after US experiments to measure drug release kinetics (37°C, either standing or shaking (95 or 180 rpm)). If the pockets exhibited MeB release within a couple of days of insonation, these pockets were also considered to be ruptured (even if no MeB was visualized immediately during insonation). These two methods of assessing US-triggered drug release were combined for analysis.

In addition, MeB release over time was measured for control (no US) and US-triggered pockets. For the controls, pockets with rupturable layers made of neat PLA or VAN-embedded PLA were incubated, without insonation, to measure passive MeB release rates. The pockets were placed in individual containers with 400 mL PBS, at 95 rpm 37°C for 7 days (Trial 0, $n = 4$ each); 1 mL drug release samples were collected daily and replaced with fresh PBS. Since the sample collected was < 1%, there was no correction made for the amount of MeB removed.

For US-triggered drug release experiments, a total of 9 pockets over three trials were tested: (1) $n = 1$, (2) $n = 4$, and (3) $n = 4$. The pockets in these trials included VAN-embedded rupturable PLA film, 10–50 mg ground VAN used in each film. In these experiments, pockets were loaded with Sonazoid microbubbles and insonated using clinical US, based on prior experiments showing equivalence of different cavitation nuclei and US methods. The pockets were insonated and incubated in containers with 400 mL PBS. The water bath during insonation for Trial #1 was at room temperature; for Trials #2 and #3, the bath was heated to 37°C. Drug release for Trial #1 was followed under static incubation whereas for trials #2 and #3, the pockets underwent shaking incubation at 95 rpm. All pockets in all trials received US on Day 0, while the pocket in Trial #1 also received a second US on Day 6. The drug release samples were collected at various time-points between Day 0 and 28.

The collected samples were stored at 4°C until the last day of collection and then measured using spectrophotometry. 200 µL of the samples were placed in Flat Bottom 96-well Microtiter plates (Fisher Scientific, Hampton, NH, USA). The concentration of MeB was measured at $\lambda = 605$ nm (Tecan Infinite M1000, Männedorf, Switzerland). A MeB standard curve was run concurrently for each trial. The total amount of MeB released as % of total MeB loaded in each pocket was calculated.

### *In vivo* rabbit pocket studies

In a pilot study, PLA film pockets assembled as above were placed into the backs of 5 male New Zealand White (NZW) rabbits (6–12 months, 4–5 kg, Charles River, Wilmington, MA, USA). The 3–4 mL pocket reservoir. contained 1–2.6 mL MeB and 1.0 mL Sonazoid microbubbles. Animal experiments were performed according to a protocol (Protocol Number: 22-11-606) approved by the Institutional Animal Care & Use Committee (IACUC) of Thomas Jefferson University and in accordance with the National Research Council's "Guide for the Care and Use of Laboratory Animals." All surgery was performed under 1–4% isoflurane anesthesia, and all efforts were made to minimize suffering.

Rabbits were pre-medicated with ketamine 30–40 mg/kg, xylazine 2–5 mg/kg, and acepromazine 0.25–1.00 mg/kg. Anesthesia was induced with 4–5% isoflurane and maintained with 1–4% isoflurane during the entire procedure. Sterile dissection was performed down to the plane between the panniculus carnosus and underlying muscle fascia. A 4 cm subcutaneous pouch was made to accommodate a pocket. For each animal, up to two pockets were placed on each side of the back, for a maximum of 4 pockets. Pockets containing MeB were either formed from neat PLA or from PLA containing embedded VAN. Of the 5 animals, one contained 2 neat PLA pockets; two contained 4 neat PLA pockets; and two contained 4 VAN-embedded PLA pockets.

Following implantation, the rabbits were allowed unrestricted ambulation and observed for activity and recovery. Wounds were inspected daily for drainage, erythema, warmth, and swelling, as well as MeB release (*i.e.*, blue staining under the wound area). After 3 days of recovery, two of the animals containing 4 pockets (one with PLA pockets and one with VAN-embedded PLA pockets) were placed under sedation and their pockets insonified for 30 minutes in power Doppler imaging mode using an S50 clinical US scanner with a curvilinear C1-6 probe and the same settings as the *in vitro* studies. Pockets were monitored for MeB release during scanning as well as daily for a total of 6 days, at which time all animals were euthanized with a barbiturate overdose in accordance with AVMA recommendations. Primary endpoints were pocket rupture, and clinical signs of distress (*i.e.*, pain, sepsis, mobility, or ability to thrive).

## Statistical analysis

US-triggered drug release outcomes were analyzed based on how much PLA was used to cast the film (with a cut-off value of 0.5 g), presence of VAN embedded in rupturable film, type of cavitation agent, or type of US modality. A total of 48 outcomes were organized as contingency tables for each variable, except for the analysis of the type of cavitation nuclei, where 45 valid outcomes were included. For comparing MeB release from insonated vs uninsonated pockets collection days 7, 8 or 6 (in that order) were included in the analysis.

Data was tested for normality with the Shapiro-Wilk test. After confirming normality, Student's t test or One-way ANOVA with Tukey's multiple comparisons test were conducted; if any of the groups tested was not normal, Mann-Whitney U test or Kruskal Wallis test with Dunn's multiple comparisons test were conducted instead. Additionally, contingency tables for US-triggered drug release outcomes were analyzed with Fisher's exact test. All of the analyses were conducted using Prism 9 (GraphPad Software, San Diego, CA; $\alpha < 0.05$ for all).

## Results

### US-triggered drug release *in vitro* depends on film thickness

The US-triggered drug release experiments were analyzed according to rupturable film thickness, which is directly correlated with the amount of PLA poured over the same surface area (Fig 3). Of a total of 48 pockets tested in US experiments, 39 were made of ≤ 0.5 g PLA rupturable film (thin rupturable film), while 9 were made of > 0.5 g PLA (thick rupturable film). In the thin film group, 21 ruptured, whereas for the thick film group, only 1 ruptured. As expected, the thinner film made the pockets more likely to undergo US-triggered rupture (Fisher's exact, $p = 0.028$).

### Rupture is not dependent on US energy or contrast agent

We next asked if increasing acoustic intensity levels beyond the clinical range, specifically through use of HIFU, increased the rupture rate of the pockets (Fig 4). Despite the increased intensity of HIFU, no differences were found ($p > 0.99$).

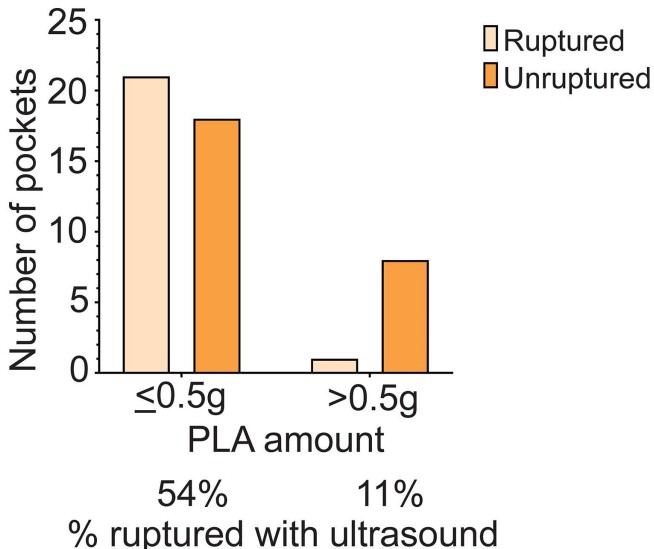

**Fig 3. US-triggered drug release (rupture) analyzed according to PLA film thickness.**

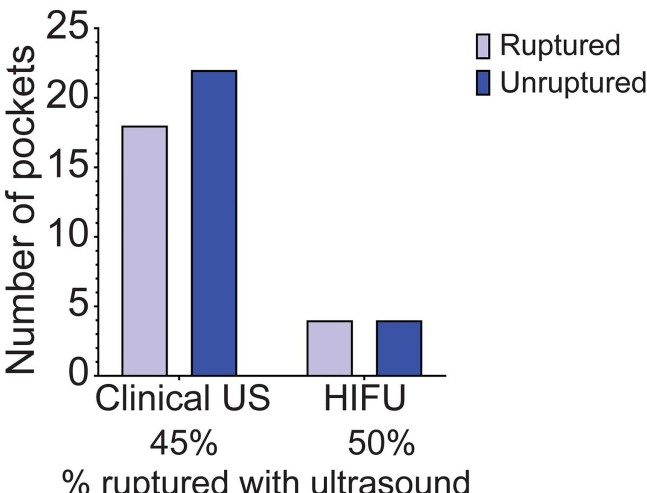

**Fig 4. US-triggered drug release (rupture) analyzed according to the type US.**

Because HIFU did not increase pocket rupture, subsequent experiments were conducted with the clinical US modes. The use of clinical US is an advantage as it allows for visualization of the pocket on the scanner and ultimately, allows more ready translation to clinical usage.

We next investigated the effect of different cavitational nuclei on microbubble-enhanced US-triggered drug release; Sonazoid microbubbles and Definity-derived acoustic nanodroplets were compared (Fig 5). Pockets containing Sonazoid microbubbles or Definity-derived acoustic nanodroplets showed no differences in terms of percentage of rupture, with ~50% of pockets ruptured in both situations ($p > 0.99$). Therefore, as the Sonazoid bubbles were active up to 2 weeks after reconstitution (Definity droplets are only acoustically active for a few hours) [26], Sonazoid was used for the remaining experiments.

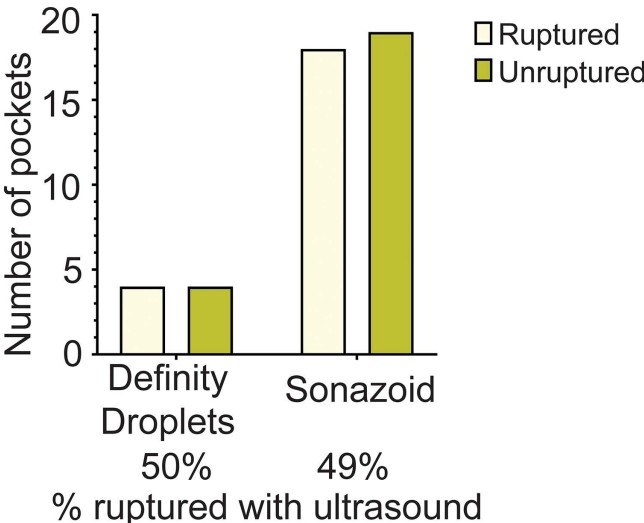

**Fig 5. US-triggered drug release (rupture) analyzed according to the type of cavitation nuclei included within the pocket, Definity-derived acoustic droplets or Sonazoid microbubbles.**

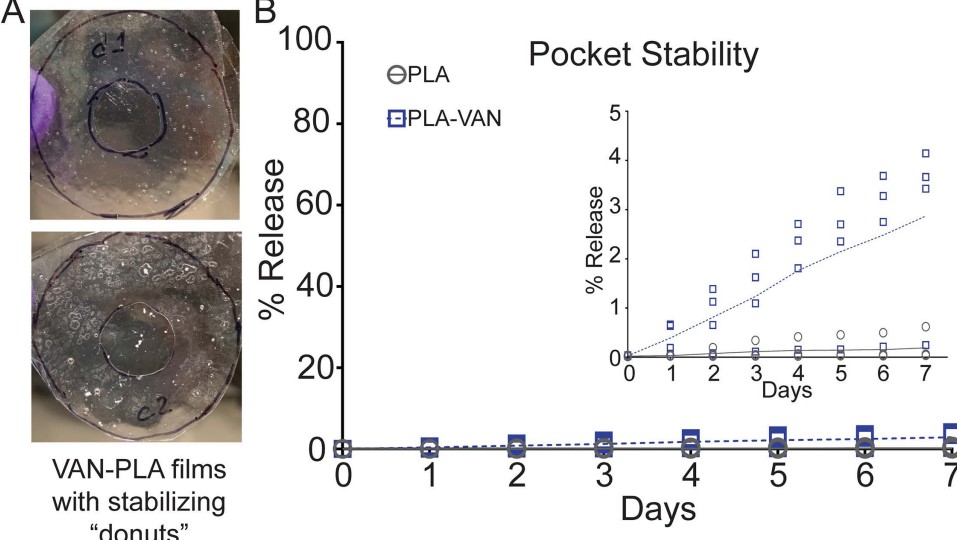

## Inclusion of VAN impurities into the thin film maintains short-term pocket stability, *in vitro*

The stability of PLA pockets containing local impurities in the film was determined with PLA pockets where VAN had been incorporated into the thin films. Time-dependent MeB release from the neat or VAN-embedded PLA pockets was measured in the presence of gentle agitation (95 rpm) (Fig 6). By Day 3, on average, the neat PLA and VAN-embedded PLA pockets released 0.11% and 1.23% of the total payload, respectively. By Day 7, the end of the experiment, these release rates had risen to 0.19% and 2.87%, on average, respectively. Overall, drug release from the two types of pockets was very low, indicating satisfactory stability. VAN-containing films were not significantly different from neat PLA films during the first 7 days of incubation (U tests, $p = 0.06$), the maximum time that pockets would reside in the wound site prior to rupture.

## VAN inclusion decreases mechanical stability/toughness

We determined stress-strain curves in PLA-VAN film pockets. 0.5 g PLA films, neat or embedded with VAN, were tested; representative curves are shown. The neat film pieces exhibited an extended plastic deformation phase compared to VAN-embedded pieces (Fig 7-A). During plastic deformation, the material undergoes changes that are not reversible; however, it does not fail completely. Notably, the ultimate strength of the neat PLA film pieces was also greater than that of VAN-embedded pieces (Fig 7-B; $19.8 \text{N/mm}^2 \pm 6.7$, $n = 10$, vs $10.3 \text{N/mm}^2 \pm 3.5$, $n = 11$; $p = 0.0011$).

The toughness of a material takes into account both the range of displacement and ultimate strength calculated for the two groups (Fig 7-C). In all, the mechanical analysis of the VAN-embedded films indicates that these films are significantly less tough (U test, $p < 0.0001$), which may make these films more susceptible to US-triggered drug release than neat PLA.

## PLA-VAN films have increased roughness

The AFM neat PLA micrographs showed local irregularities due to miniscule air pockets and dust particles (Fig 8-A). These inclusions were apparent on the 3D maps although their effect on overall smoothness was small. The PLA-VAN film (Fig 8-B) showed an overall smooth surface interrupted by inclusions of various sizes and distribution. In the largest inclusions, granulated material consistent with the presence of VAN were observed. The topographic map of the PLA-VAN

**Fig 6. Initial investigation of PLA-VAN films. A.** PLA films with incorporated VAN; **B.** MeB release over the course of a week for pockets made with neat PLA and VAN-embedded PLA ($n = 4$), incubated at 37°C, 95 rpm.

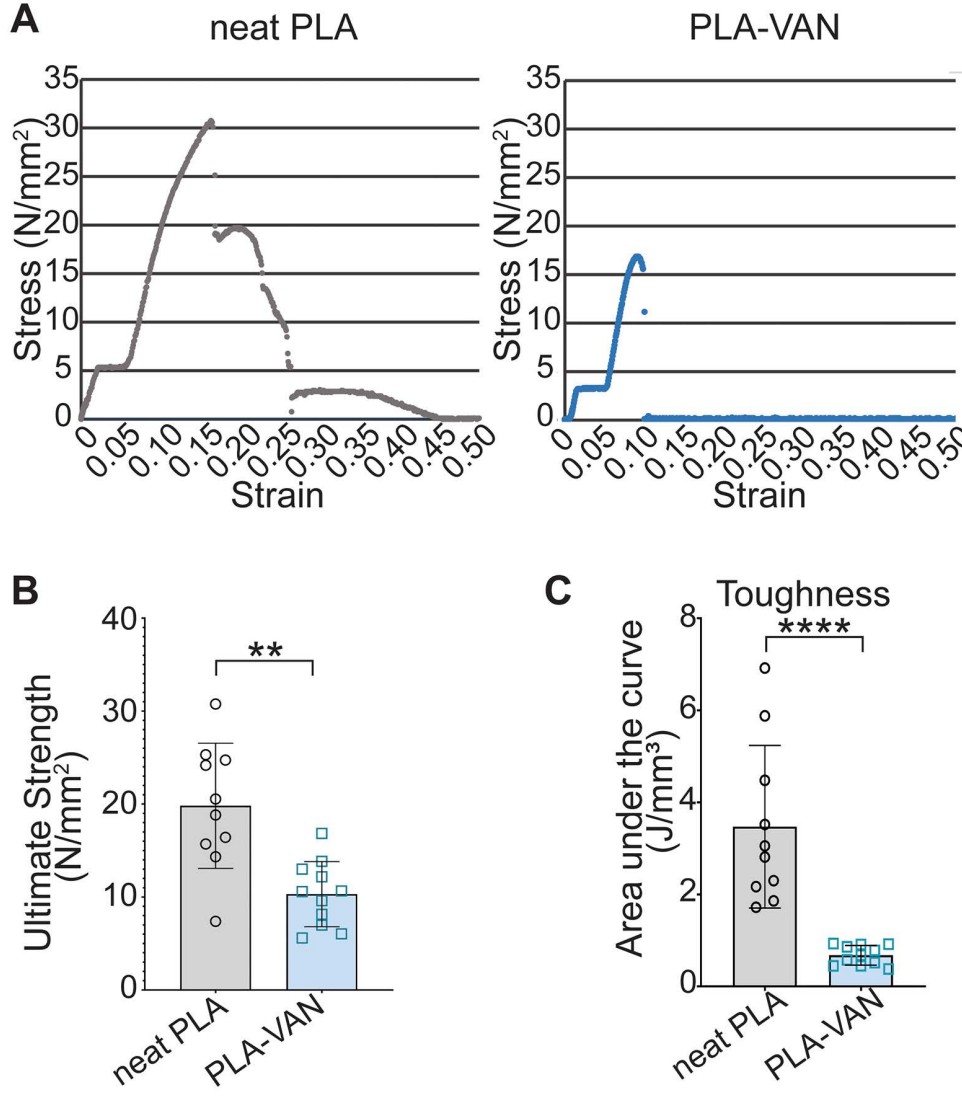

**Fig 7. Stress vs. strain and toughness of thin films. (A)** Representative stress-strain curves from neat PLA and PLA-VAN films (0.5 g). Tracings for remaining films are in the supplement (S1 Fig) **(B)** The ultimate strength of neat PLA ($n = 10$) and PLA-VAN ($n = 11$) films **(C)** The toughness of neat PLA ($n = 10$) and PLA-VAN ($n = 11$) films.

was irregular where the frequency of the deviations from the surface increased suggesting a film that was affected by the VAN. The surface topography of the neat PLA film had an RMS roughness of $16.00 \pm 5.75$ nm, while the PLA-VAN film had an RMS roughness of $38.24 \pm 12.77$ nm ($p = 0.0192$). These data suggest that rupture can be achieved more easily with VAN-PLA films, which is beneficial for drug release. Additionally, this finding is consistent with data from the previously discussed mechanical testing.

## PLA-VAN films show increased US rupture

Because the PLA-VAN films were more brittle than neat PLA films, we predicted that these would be more readily ruptured with US. US treatment caused drug release in 92% of the pockets with films containing VAN, indicating enhanced suscep-tibility to acoustic pressure. Only 31% of the pockets with neat PLA films were ruptured (Fig 9; $p = 0.0004$).

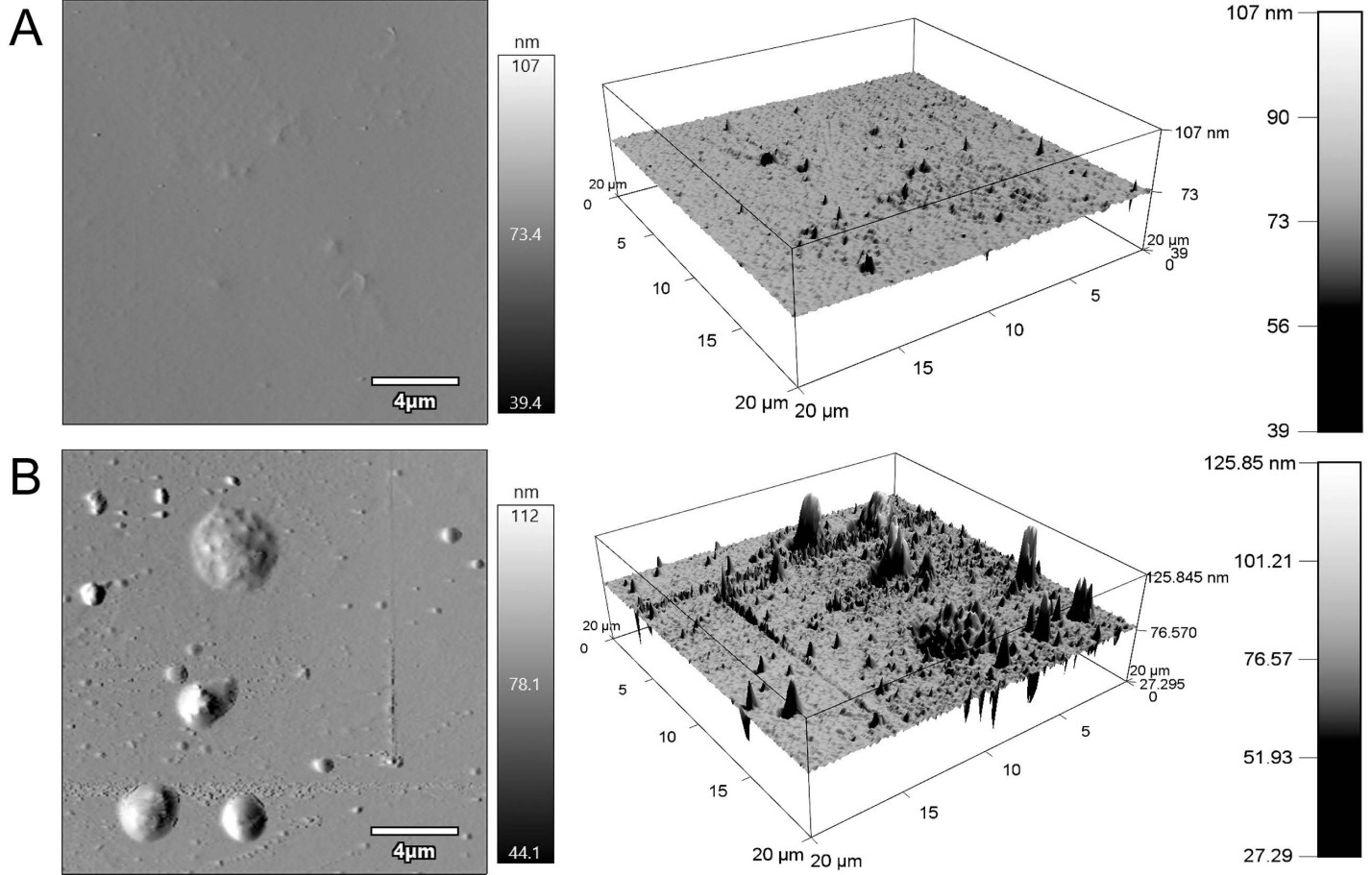

**Fig 8. Amplitude retrace topography images and 3D surface mapping of PLA films captured over a 20 x 20 μm area. (A)** Surface characteristics of neat PLA film exhibiting consistent, smooth surface topography with minimal irregularities due to air pockets or dust particles. **(B)** PLA-VAN film with topographic map revealing distinct VAN granulations consistently across the film's surface with varying concentration and distribution.

Next, US-triggered MeB release from VAN-embedded pockets ($n = 9$) were measured. Eight of the nine pockets (88.9%) were successfully ruptured with US and exhibited approximately linear release (best fit is indicated by dashed line, Fig 10). One pocket, indicated with "X's", had a rapid post-insonation release that plateaued around 55% with release kinetics that were sigmoidal.

MeB release from uninsonated PLA-VAN pockets was $2.87 \pm 1.77\%$ (Fig 4-B); average release from insonated, VAN-embedded, PLA film pockets was $10.57 \pm 17.29\%$. Overall, variability was large and the difference between release rates was not statistically significant (U test, $p = 0.50$).

### VAN is immediately released from PLA-VAN films upon aqueous submersion

As pockets with VAN-embedded rupturable films showed the most reproducible outcomes for US-triggered drug release, we studied if the film-associated VAN, which would be eluted upon placement of the pockets, was sufficient to afford a secondary antibiotic prophylaxis in the local surgical wound site. For a first trial, the percent VAN release from the film in the first day and up to 15 days (Fig 11-A) are shown. Total release values are close to 2.5 mg or 2,500 μg ($n = 3$; $p = 0.19$).

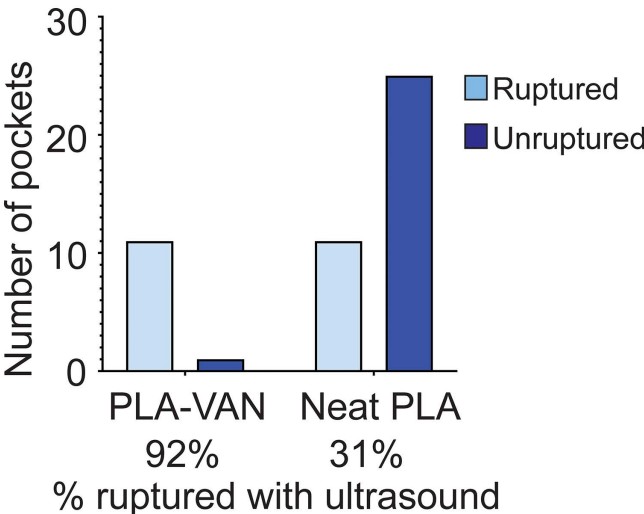

**Fig 9. US-triggered drug release (rupture) analyzed according to the presence of VAN in the rupturable PLA film.**

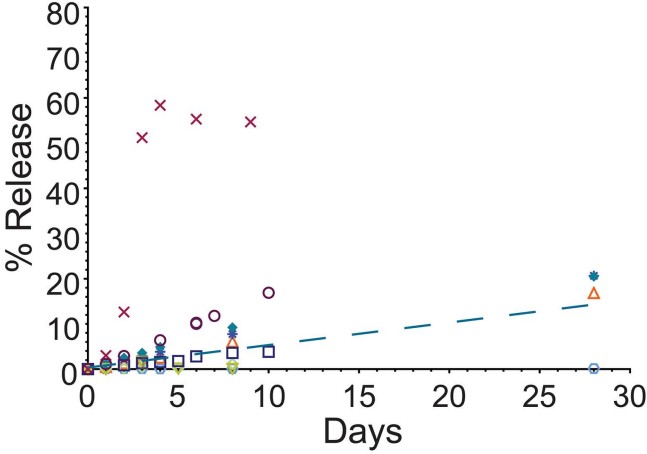

**Fig 10. MeB drug release over time for insonated PLA-VAN pockets.** Separate pockets are indicated with different symbols/colors. The best fit to the data by linear regression (the trial indicated with X's was excluded from this calculation) is shown with the blue dashed line, with an $R^2 = 0.59$. US was applied at room temperature and drug release followed with incubation (37°C). ($n = 1$-9/individual time).

Despite the variability in each sample, superficial VAN dissolved in the first couple of hours, in keeping with many passive elution systems. We next investigated if VAN release was still present at 27 days. Again, we measured immediate VAN release expressed as percent of the total loading ($n = 3$) on the first day and up to 27 days, which was statistically significantly higher ($p = 0.0436$; Fig 11-B). Most of the VAN dissolved early in the incubation period, although some additional release was measured after the first day, which we speculate is due to some PLA degradation which releases additional VAN.

In summary, the *in vitro* experiments established a design for an US-rupturable pocket that could release drug over time, ultimately towards establishing a physician-triggered release system.

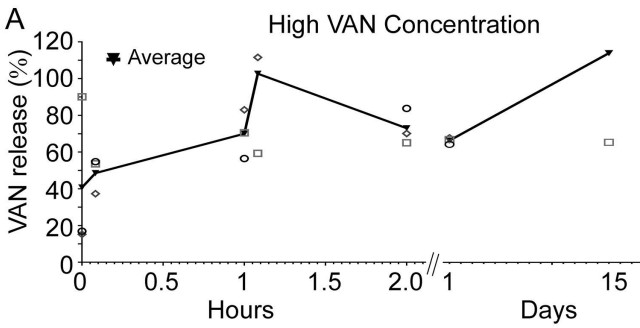

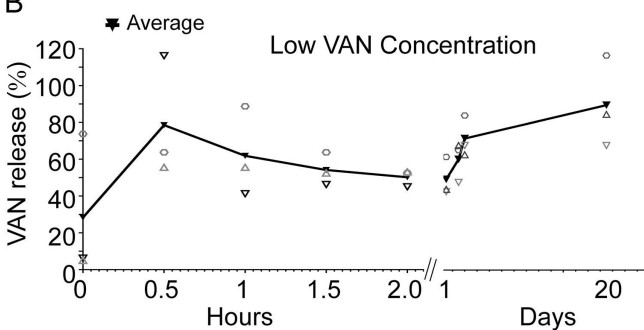

**Fig 11.** Relative VAN release from VAN-embedded 0.50 g PLA film pieces. Trial 1, high concentration VAN **(A)**, and Trial 2, low concentration VAN **(B)**, over time.

### Both neat PLA and PLA-VAN pockets show US-induced MeB release *in vivo*

Finally, we tested the pockets *in vivo* by inserting MeB-containing pockets in the back of rabbits next to the spine. Blue staining could be seen through the rabbit skin (Fig 12) allowing assessment of rupture of the pockets, both before (indicating lack of stability) and after US insonation.

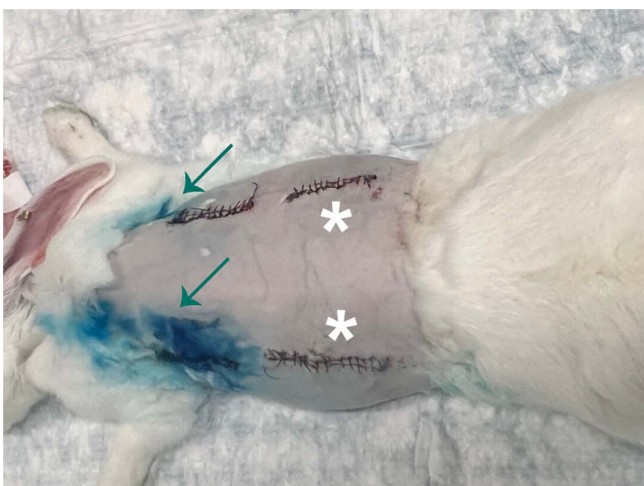

**Fig 12.** Representative rabbit (#3), imaged on Day 3 prior to US. The superior pockets demonstrate blue dye (MeB) leak (arrows), whereas the inferior sites (stars) are dry and did not show a leak.

For the control (uninsonated) pockets, 4 out of 8 (50%) were intact by day 3, while 5 out of the 7 pockets (71%) assigned to US insonation were intact by day 3 resulting in an overall rate of stability of 60% (9/15) by day 3 (Table 1). Importantly, all 5 of the intact pockets (3 neat PLA and 2 PLA-VAN) that were allocated for insonation ruptured following US (100%; p > 0.99). For the neat PLA pockets, 5 out of 9 were intact on day 3, while 4 out of 6 PLA-VAN pockets survived until day 3 (p > 0.99). An additional pocket from Rabbit #5 was excluded from analysis, due to the emptying of the pocket contents occurring without staining, making it impossible to determine the time-line of leakage.

## Discussion

This study introduced a new type of local drug delivery system that retains the payload until triggered by application of US. This triggering results in an immediate delivery of drugs for antibiotic prophylaxis followed by a more long-term steady release (*cf.*, Fig 10) of drugs for antibiotic prophylaxis. We created a system for use in instrumented spine surgery that could be activated by US on a delayed timeline, *i.e.*, after the surgical drains are removed and VAN powder has dissipated. PLA films were used to construct a drug-loaded (in these experiments VAN) pocket containing up to 4.5 mL. Since 1 g of VAN is soluble in 3.0 mL of water a single pocket can replicate a wide range of concentrations including that of the initial intraoperative antibiotics. The ability of our device to be tuned to the requirements of the individual patient is an important feature.

The leading methods for prevention and treatment of implant-associated infections with local antibiotics are topical antibiotic powder (usually VAN), antibiotic-infused polymethylmethacrylate cement, and biodegradable antibiotic carriers, such as calcium sulfate [27]. Antibiotic-loaded cement spacers are commonly used in periprosthetic joint infections or 1- or 2-stage exchange arthroplasty [27]; however, for spinal surgery, none of the local antibiotic delivery methods has garnered enough evidence to become the standard of care. VAN powder has been employed and studied the most for prevention of SSIs after instrumented spine surgery [28,29]; where placing free VAN powder in the surgical space results in very high initial levels followed by rapid depletion of antibiotic concentrations over time [30]. Biodegradable carriers offer another option and aim to release antibiotics; however, they too suffer from burst release and much of recent research has focused on how to achieve controlled release [11,27].

To achieve triggered release, we focused on PLA as it is the leading biocompatible and biodegradable polymer in clinical application design, including orthopaedics [31–35]. Examples include biodegradable sutures, orthopaedic biodegradable screws and plates, and after spinal surgeries including laminectomy [31,32]. Similar to hydrogels, PLA constructs can be fine-tuned for the needs of drug delivery applications with alterations in mechanical strength and degradation kinetics [33]. Upon degradation into smaller monomers and oligomers, PLA is assumed to be excreted from the body via urination or exhalation. As such, additional surgeries to remove the PLA device are not necessary, improving patient recovery and optimizing health system costs [31,32]. We previously showed that thin PLA films on a spinal clip would rupture with US to

**Table 1. Experimental setup of pockets and rabbits with stability and US-triggered drug release outcomes.**

| Rabbit # | # of pockets | Pocket composition | Day 3 Intact | % of intact pockets ruptured by US |
|---|---|---|---|---|
| 1 Control | 4 | Neat PLA | 1/4 | n/a |
| 4 Control | 3 | VAN-embedded PLA | 2/3 | n/a |
| 5 Control | 1* | Neat PLA | 1/1 | n/a |
| 2 Active US | 4 | Neat PLA | 3/4 | 100% (3 of 3) |
| 3 Active US | 3 | VAN-embedded PLA | 2/3 | 100% (2 of 2) |
| Total | 15 | | 9/15 | |

VAN: Vancomycin, MeB: methylene blue dye. *An additional pocket from Rabbit #5 was excluded from analysis for experimental reasons.

trigger bulk release [11]. Unlike many other studies that rely on PLA, or PLA combinations [35] or controlled porosity [31], in our design, the release of the drug payload is instead achieved with US. The final design (a thick, base layer film and a thin/donut-stabilized rupturable layer film) consists entirely of a PLA film exterior that confers a favorable biosafety profile, because it will eventually degrade and be naturally eliminated from the body.

Application of US, a noninvasive, relatively inexpensive technique with high spatial and temporal resolution, is an attractive method for controlled drug delivery [18]. We hypothesized that the higher power of HIFU may prove more successful in terms of drug delivery, but clinical US and HIFU results were comparable; clinical US was ultimately preferred for further study due to its higher translational potential. We turned to the use of cavitation agents, microbubbles and nanodroplets, to achieve a higher acoustic effects while keeping the insonation intensity low and within clinical limits [36]. Bubble-enhanced drug delivery has been explored in scaffolds and against biofilms; bubbles have also been used as drug delivery vehicles themselves [16,37,38].

The final design consisted of a thick, base layer film and a thin, rupturable layer film; the rupturable component was dual-layer, with a thick donut piece surrounding a thin film center. The use of US also augments our antibacterial strategy [39,40]. US may disrupt the organization of sessile bacteria in biofilms, enhance permeation of antibiotics through biofilm layers, and increase the susceptibility of bacteria to lower doses of antibiotics [17]. Many different US parameters have been tested previously for US-controlled drug delivery, as well as the inclusion cavitation nuclei. These gas-filled cavitation agents undergo phase change in response to the acoustic energy, thereby having disruptive effects on the local surroundings [18]. We tested Sonazoid microbubbles and Definity acoustic droplets, expecting an enhancement of drug delivery with Definity droplets since the phase change of these droplets was more remarkable than for Sonazoid microbubbles. However, the results were comparable and, again, Sonazoid microbubbles were preferred for later stages of investigation due to previous approvals [36].

Our most successful design incorporated a thin PLA film with embedded VAN powder sealed onto a thick film cone. The PLA-VAN films had a lower ultimate strength and toughness than neat PLA counterparts and achieved more predictable rupture. The inclusion of VAN in our PLA films resulted in elution of VAN into the surroundings, even in the absence of US. In the presence of existing peri-operative prophylaxis with VAN powder, the benefit of additional VAN in solution is likely to be negligible, but its presence on the PLA may provide an important layer of protection against colonization of the immediate PLA surface [41].

Even though the PLA film pockets achieved US-triggered drug release *in vitro*, the pocket design did not meet all of our initial criteria. We aimed for bolus release but achieved sustained release. While extended antibiotic prophylaxis has shown some benefits in high risk primary joint surgeries and revisions [42], guidelines state that extended antibiotics after surgery for prevention of infection is not beneficial [43,44], even in riskier trauma surgeries, such as open fractures [45]. Our system achieved a release that is more appropriate for situations which require noninvasive, controlled, sustained drug delivery that can overcome the burst release phenomenon, such as treatment of orthopaedic infections, which require extended courses of antibiotics [46], or long-term hormonal contraception [34], which may benefit from the current release characteristics of this pocket design.

*In vivo*, bolus release was achieved in our pilot rabbit study. The *in vivo* studies showed that the current design had varying degrees of stability, with 4/8 of control pockets and 5/7 active pockets remaining intact during the 3-day waiting period. Stability of PLA and PLA-VAN films were equivalent (p > 0.99); albeit based on a very small sample size. When the pockets were insonated by US, all pockets that survived the 3-day implantation (5 in total) ruptured. We suggest that manipulation of this stability could either bias the pockets towards limited release during early implantation or no release; both sets could then achieve rupture and rapid drug release with US treatment.

Throughout, the casting of the PLA, due to the presence of chloroform, was considered sterile. Common methods of sterilization such as steam or ethylene oxide are difficult with PLA due to the thermal and hydrolytical sensitivity of PLA [47]. Ivanova et al. used steam sterilization on thin PLA films, which changed mechanical properties such as roughness

and hydrophilicity [48]. Assembling the PLA film pockets with sterile technique and then using ultraviolet (UV) radiation or hydrogen peroxide plasma may be more appropriate [49].

There were limitations to our studies, where additional experiments could discern differences in outcomes with regards to type of US or cavitation nuclei, and allow us to better characterize drug release profile. Another limitation to this study was the small sample size for the *in vivo* evaluations. In accordance with the 3 Rs of responsible *in vivo* research, we performed this pilot study using rabbits based on the results of our previous studies [26]. Most importantly, the stability and drug release of the *in vitro* experiments did not reflect the outcome achieved in the *in vivo* experiments. These differences could be due to the insertion environment (proteins, ions, pH), or the mechanical constraint in the insertion site combined with the pocket design. Future iterations of pocket designs would benefit from earlier and larger *in vivo* studies, where the design parameters could be adjusted (*e.g.*, not cone shaped and/or altering the hydrophilicity of the thin film PLA by replacing it with hydrophobic PLGA) according to *in vivo* outcomes. Finally, a crucial next step will be to evaluate the efficacy of this drug delivery concept against common bacteria such as *Staphylococcus aureus* similar to our recent work [50].

## Conclusions

A novel US-triggered drug delivery system based on PLA film pockets was developed and tested. VAN-embedded PLA film pockets were most successful at achieving US-enhanced drug release *in vitro*, demonstrating a linear drug release for the few weeks following insonation in most pocket experiments. This design can be utilized for the treatment of orthopaedic infections, delivering a high concentration of antibiotics locally in a sustained fashion. Further research may be required to make the PLA film pockets more appropriate for US-triggered bolus drug release for prophylaxis.

## Supporting information

**S1 Fig. Additional data for stress vs. strain of different thin films.** (A) curves from neat PLA films, (B) curves from PLA-VAN films.
(PDF)

**S1 Data. Study data: Data generated and/or analyzed during this study.**
(PDF)

## Acknowledgments

We thank Hebah Falatah for Definity droplet preparation and Catherine Gurr for help with data presentation.

## Author contributions

**Conceptualization:** Noreen J Hickok, Flemming Forsberg.

**Data curation:** Selin Isguven Billmyer, Ryan E Tomlinson, Lauren J Delaney, Alexander H Harris, Eric McLaughlin, Noreen J Hickok.

**Formal analysis:** Selin Isguven Billmyer, Ryan E Tomlinson, Noreen J Hickok, Flemming Forsberg.

**Funding acquisition:** Noreen J Hickok, Flemming Forsberg.

**Investigation:** Selin Isguven Billmyer, Priscilla Machado, Ryan E Tomlinson, Lauren J Delaney, Ji-Bin Liu, Alexander H Harris, Eric McLaughlin, Noreen J Hickok, Flemming Forsberg.

**Methodology:** Selin Isguven Billmyer, Priscilla Machado, Ryan E Tomlinson, Lauren J Delaney, Ji-Bin Liu, Noreen J Hickok.

**Project administration:** Noreen J Hickok, Flemming Forsberg.

**Supervision:** Noreen J Hickok, Flemming Forsberg.

**Validation:** Lauren J Delaney, Flemming Forsberg.

**Visualization:** Selin Isguven Billmyer, Ji-Bin Liu, Alexander H Harris, Eric McLaughlin, Noreen J Hickok.

**Writing – original draft:** Selin Isguven Billmyer.

**Writing – review & editing:** Selin Isguven Billmyer, Priscilla Machado, Ryan E Tomlinson, Lauren J Delaney, Ji-Bin Liu, Alexander H Harris, Eric McLaughlin, Noreen J Hickok, Flemming Forsberg.

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
