## [Decision Letter · Decision Letter 0]

28 Apr 2025

Dear Dr. Forsberg,

Thank you for submitting your manuscript to PLOS ONE. After careful consideration, we feel that it has merit but does not fully meet PLOS ONE’s publication criteria as it currently stands. Therefore, we invite you to submit a revised version of the manuscript that addresses the points raised during the review process.

We look forward to receiving your revised manuscript.

Kind regards,

Dola Sundeep

Academic Editor

PLOS ONE

“NIH R01 AR069119

The Mullin Fund for Spinal Research at Thomas Jefferson University”

“We thank Hebah Falatah for Definity droplet preparation and Catherine Gurr for help with data presentation. Research reported in this manuscript was supported by the National Institute of Arthritis and Musculoskeletal Diseases of the National Institutes of Health under award number R01AR069119 (Hickok, Forsberg, Isguven) and by the Mullin Fund for Spinal Research at Thomas Jefferson University. The content is solely the responsibility of the authors and does not necessarily represent the official views of the National Institutes of Health. “

“NIH R01 AR069119

The Mullin Fund for Spinal Research at Thomas Jefferson University”

“I have read the journal's policy and the authors of this manuscript have the following competing interests: Selin Isguven Billmyer, Noreen Hickok, Flemming Forsberg reports financial support was provided by National Institutes of Health. Flemming Forsberg reports a relationship with GE HealthCare that includes: consulting or advisory. Flemming Forsberg reports a relationship with Lantheus Medical Imaging Inc that includes: consulting or advisory. Selin Isguven Billmyert, Noreen Hickok, Flemming Forsberg has patent PCT Application No. PCT/US2023/077854 pending to Thomas Jefferson University. The other authors, declare that they have no known competing financial interests or personal relationships that could have appeared to influence the work reported in this paper.”

6. In this instance it seems there may be acceptable restrictions in place that prevent the public sharing of your minimal data. However, in line with our goal of ensuring long-term data availability to all interested researchers, PLOS’ Data Policy states that authors cannot be the sole named individuals responsible for ensuring data access (http://journals.plos.org/plosone/s/data-availability#loc-acceptable-data-sharing-methods).

Additional Editor Comments:

Subject: Decision on Your Manuscript PONE-D-25-14645

Dear Dr. Forsberg,

We have now received the required number of reviewer reports for your manuscript titled "Delayed delivery of antibiotics by ultrasound-mediated rupture of polylactic acid pockets: in vitro and in vivo studies" (Manuscript Number: PONE-D-25-14645).

Based on the reviewers' comments, the decision on your manuscript is Minor Revision.

We look forward to receiving your revised submission.

Thank you for your continued interest in our journal.

Best regards,

Dr. Dola Sundeep

Reviewers' comments:

Reviewer's Responses to Questions

**Comments to the Author**

1. Is the manuscript technically sound, and do the data support the conclusions?

Reviewer #1: Yes

Reviewer #2: Yes

2. Has the statistical analysis been performed appropriately and rigorously?

Reviewer #1: Yes

Reviewer #2: Yes

3. Have the authors made all data underlying the findings in their manuscript fully available?

Reviewer #1: Yes

Reviewer #2: Yes

4. Is the manuscript presented in an intelligible fashion and written in standard English?

Reviewer #1: Yes

Reviewer #2: Yes

Reviewer #1: This study introduces a promising and innovative ultrasound-triggered antibiotic delivery system for spinal surgery infections. The approach is highly relevant and potentially impactful, offering a non-invasive method for localized infection control. However, to strengthen the manuscript, several areas need improvement:

Statistical Analysis – Key outcomes like rupture rates and antibiotic release should include significance testing.

Long-Term Efficacy – Additional data on stability and prolonged effectiveness are needed.

Comparative Controls – Comparing with other delivery systems (e.g., beads, hydrogels) would add context.

Tissue Compatibility – More information on inflammatory responses or biocompatibility is recommended.

Ultrasound Parameters – Clearer details (e.g., frequency, intensity, duration) would aid reproducibility.

Limitations & Future Work – The authors should discuss the small sample size and outline next steps.

Overall, this is a well-designed, novel study with strong potential, but a few enhancements would improve its clarity, reproducibility, and clinical applicability.

Reviewer #2: PONE-D-25-14645 Review

I would recommend publishing however, authors should explain the points below and consider the following revisions.

1) Authors should explain why the cone shaped geometry was selected for this application.

2) Authors should consider and report MeB and VAN release data as % release / surface area of the pocket

3) How many pockets will be needed to deliver the typical bolus dose of VAN post-surgery ?

4) Have the authors altering the hydrophilicity of PLA , by replacing it with hydrophobic PLGA in the thin film ?

5) Detailed surface characterization using AFM and visualization using SEM, might help explain the difference between invitro and invivo disconnect that the authors observed.

**Do you want your identity to be public for this peer review?** For information about this choice, including consent withdrawal, please see our Privacy Policy

Reviewer #1: **Yes: ** Nour H. Aboalhaija

Reviewer #2: No

---

## [Author Response · Author response to Decision Letter 1]

13 Oct 2025

August 12, 2025

Dola Sundeep

Academic Editor

PLOS ONE

Re: Delayed delivery of antibiotics by ultrasound-mediated rupture of polylactic acid pockets: in vitro and in vivo studies; Manuscript no: PONE-D-25-14645

Dear Dr. Sundeep

Thank you for your email of April 28th, 2025 and for the reviewer’s insightful comments on our manuscript referenced above. Our responses to the questions raised can be found below. All line references refer to the revised and marked version of the manuscript. Finally, we discovered a few trivial typographical and linguistic errors in the original manuscript and those have also been corrected.

We have carefully gone through the entire manuscript and made sure it conforms to the journal guidelines.

As requested we have expanded the information regarding our animal experiments, which now reads (in part):

“Animal experiments were performed according to a protocol (Protocol Number: 22-11-606) approved by the Institutional Animal Care & Use Committee (IACUC) of Thomas Jefferson University and in accordance with the National Research Council's “Guide for the Care and Use of Laboratory Animals.” All surgery was performed under 1-4% isoflurane anesthesia, and all efforts were made to minimize suffering.

Rabbits were pre-medicated with ketamine 30-40 mg/kg, xylazine 2-5 mg/kg, and acepromazine 0.25-1.00 mg/kg. Anesthesia was induced with 4-5% isoflurane and maintained with 1-4% isoflurane during the entire procedure. ...

Following implantation, the rabbits were allowed unrestricted ambulation and observed for activity and recovery. Wounds were inspected daily for drainage, erythema, warmth, and swelling, as well as MeB release … for a total of 6 days, at which time all animals were euthanized with a barbiturate overdose in accordance with AVMA recommendations.” (lines 247 - 268)

“NIH R01 AR069119

The Mullin Fund for Spinal Research at Thomas Jefferson University”

Thank you for your guidance. The cover letter now includes the statement:

“NIH R01 AR069119

The Mullin Fund for Spinal Research at Thomas Jefferson University

“We thank Hebah Falatah for Definity droplet preparation and Catherine Gurr for help with data presentation. Research reported in this manuscript was supported by the National Institute of Arthritis and Musculoskeletal Diseases of the National Institutes of Health under award number R01AR069119 (Hickok, Forsberg, Isguven) and by the Mullin Fund for Spinal Research at Thomas Jefferson University. The content is solely the responsibility of the authors and does not necessarily represent the official views of the National Institutes of Health. “

“NIH R01 AR069119

The Mullin Fund for Spinal Research at Thomas Jefferson University”

Thank you for your guidance. The cover letter now includes the statement:

“NIH R01 AR069119

The Mullin Fund for Spinal Research at Thomas Jefferson University

Moreover, we have removed the funding information from the Acknowledgement section, which now reads”

“We thank Hebah Falatah for Definity droplet preparation and Catherine Gurr for help with data presentation.” (lines 561 - 562)

“I have read the journal's policy and the authors of this manuscript have the following competing interests: Selin Isguven Billmyer, Noreen Hickok, Flemming Forsberg reports financial support was provided by National Institutes of Health. Flemming Forsberg reports a relationship with GE HealthCare that includes: consulting or advisory. Flemming Forsberg reports a relationship with Lantheus Medical Imaging Inc that includes: consulting or advisory. Selin Isguven Billmyert, Noreen Hickok, Flemming Forsberg has patent PCT Application No. PCT/US2023/077854 pending to Thomas Jefferson University. The other authors, declare that they have no known competing financial interests or personal relationships that could have appeared to influence the work reported in this paper.”

Please confirm that this does not alter your adherence to all PLOS ONE policies on sharing data and materials, by including the following statement: "This does not alter our adherence to PLOSONE policies on sharing data and materials.” (as detailed online in our guide for authors http://journals.plos.org/plosone/s/competing-interests). If there are restrictions on sharing of data and/or materials, please state these. Please note that we cannot proceed with consideration of your article until this information has been declared.

Thank you for your guidance. The cover letter now includes the statement:

“I have read the journal's policy and the authors of this manuscript have the following competing interests: Selin Isguven Billmyer, Noreen Hickok, Flemming Forsberg report financial support was provided by National Institutes of Health. Flemming Forsberg reports a relationship with GE HealthCare that includes: consulting or advisory. Flemming Forsberg reports a relationship with Lantheus Medical Imaging Inc that includes: consulting or advisory. Selin Isguven Billmyer, Noreen Hickok, Flemming Forsberg have patent PCT Application No. PCT/US2023/077854 pending to Thomas Jefferson University. The other authors declare that they have no known competing financial interests or personal relationships that could have appeared to influence the work reported in this paper. This does not alter our adherence to PLOS One policies on sharing data and materials.”

6. In this instance it seems there may be acceptable restrictions in place that prevent the public sharing of your minimal data. However, in line with our goal of ensuring long-term data availability to all interested researchers, PLOS’ Data Policy states that authors cannot be the sole named individuals responsible for ensuring data access (http://journals.plos.org/plosone/s/data-availability#loc-acceptable-data-sharing-methods).

Our institution does not have a data access committee or anything similar. We will provide a complete copy of all the data to Dr. John Eisenbrey (Professor of Radiology) who will maintain long term availability of the data from this study and who will handle any requests regarding access. His email is: john.eisenbrey@jefferson.edu

We are not aware of any retractions among our reference list have updated the list to reflect the PLOS One style, as requested. Moreover, we replaced the abstract cited as (26) with the peer-reviewed paper of this study. Finally, three new references were added to comply with the expanded discussion requested by the reviewers (32, 33, 50).

Additional Editor Comments:

Subject: Decision on Your Manuscript PONE-D-25-14645

Dear Dr. Forsberg,

We have now received the required number of reviewer reports for your manuscript titled "Delayed delivery of antibiotics by ultrasound-mediated rupture of polylactic acid pockets: in vitro and in vivo studies" (Manuscript Number: PONE-D-25-14645).

Based on the reviewers' comments, the decision on your manuscript is Minor Revision.

We look forward to receiving your revised submission.

Thank you for your continued interest in our journal.

Best regards,

Dr. Dola Sundeep

Reviewers' comments:

Reviewer's Responses to Questions

Comments to the Author

1. Is the manuscript technically sound, and do the data support the conclusions?

Reviewer #1: Yes

Reviewer #2: Yes

2. Has the statistical analysis been performed appropriately and rigorously?

Reviewer #1: Yes

Reviewer #2: Yes

3. Have the authors made all data underlying the findings in their manuscript fully available?

Reviewer #1: Yes

Reviewer #2: Yes

4. Is the manuscript presented in an intelligible fashion and written in standard English?

Reviewer #1: Yes

Reviewer #2: Yes

5. Review Comments to the Author

Reviewer #1: This study introduces a promising and innovative ultrasound-triggered antibiotic delivery system for spinal surgery infections. The approach is highly relevant and potentially impactful, offering a non-invasive method for localized infection control. However, to strengthen the manuscript, several areas need improvement:

Statistical Analysis – Key outcomes like rupture rates and antibiotic release should include significance testing.

We tested the rupture rates for neat PLA and PLA-VAN pockets and found no statistically significant difference (p > 0.99). We also clarified the issue with the one excluded pocket and the text now reads:

“Importantly, all 5 of the intact pockets (3 neat PLA and 2 PLA-VAN) that were allocated for insonation ruptured following US (100%; p > 0.99). For the neat PLA pockets, 5 out of 9 were intact on day 3, while 4 out of 6 PLA-VAN pockets survived until day 3 (p > 0.99). An additional pocket from Rabbit #5 was excluded from analysis, due to the emptying of the pocket contents occurring without staining, making it impossible to determine the timeline of leakage.” (lines 433 - 438).

The antibiotic release rates were converted to % (as requested by Reviewer 2) and testing showed no increase by day 15 (p = 0.19) but after 27 days the release was statistically significantly higher (p = 0.0436). Both p-values have now been added to the text, which was rewritten to:

” For a first trial, the percent VAN release from the film in the first day and up to 15 days (Fig. 11-A) are shown. Total release values are close to 2.5 mg or 2,500 µg (n = 3; p = 0.19).

Despite the variability in each sample, superficial VAN dissolved in the first couple of hours, in keeping with many passive elution systems. We next investigated if VAN release was still present at 27 days. Again, we measured immediate VAN release expressed as percent of the total loading (n = 3) on the first day and up to 27 days, which was statistically significantly higher (p = 0.0436; Fig. 11-B). Most of the VAN dissolved early in the incubation period, although some additional release was measured after the first day, which we speculate is due to some PLA degradation which releases additional VAN.“ (lines 403 - 416).

Long-Term Efficacy – Additional data on stability and prolonged effectiveness are needed.

The premise of our local drug delivery system is that all pockets must be activated within 6 days to reduce the possibility of bacterial infections taking hold (i.e., for prophylaxis). There is therefore no need for long-term efficacy studies, and we have not made any changes to the manuscript in this regard.

Comparative Controls – Comparing with other delivery systems (e.g., beads, hydrogels) would add context.

While we do agree with the reviewer, we feel that such experiments are outside the scope of this manuscript. We do ho

---

## [Decision Letter · Decision Letter 1]

12 Nov 2025

Delayed delivery of antibiotics by ultrasound-mediated rupture of polylactic acid pockets: in vitro and in vivo studies

PONE-D-25-14645R1

Dear Dr. Flemming Forsberg,

We’re pleased to inform you that your manuscript has been judged scientifically suitable for publication and will be formally accepted for publication once it meets all outstanding technical requirements.

Kind regards,

Dola Sundeep

Academic Editor

PLOS ONE

Additional Editor Comments (optional):

Dear Dr. Flemming Forsberg,

Thank you for submitting your manuscript entitled “Delayed Delivery of Antibiotics by Ultrasound-Mediated Rupture of Polylactic Acid Pockets: In Vitro and In Vivo Studies” (Manuscript ID: PONE-D-25-14645R1) to PLOS ONE.

Based on the reviewers’ comments and the satisfactory revisions made in response, I am pleased to inform you that the final editorial decision is to Accept your manuscript for publication in PLOS ONE.

We appreciate your careful attention to the reviewers’ suggestions and the improvements made to enhance the quality and clarity of your work. Congratulations on this achievement, and thank you for choosing PLOS ONE as the venue for your research.

Kind regards,

Dr. Dola Sundeep

Academic Editor

PLOS ONE

Reviewers' comments:

Reviewer's Responses to Questions

**Comments to the Author**

Reviewer #2: All comments have been addressed

2. Is the manuscript technically sound, and do the data support the conclusions?

Reviewer #2: Yes

3. Has the statistical analysis been performed appropriately and rigorously?

Reviewer #2: Yes

4. Have the authors made all data underlying the findings in their manuscript fully available?

Reviewer #2: Yes

5. Is the manuscript presented in an intelligible fashion and written in standard English?

Reviewer #2: Yes

Reviewer #2: (No Response)

**Do you want your identity to be public for this peer review?** For information about this choice, including consent withdrawal, please see our Privacy Policy

Reviewer #2: No

---

## [Editor Report · Acceptance letter]

PONE-D-25-14645R1

PLOS ONE

Dear Dr. Forsberg,

I'm pleased to inform you that your manuscript has been deemed suitable for publication in PLOS ONE. Congratulations! Your manuscript is now being handed over to our production team.

Kind regards,

on behalf of

Dr. Dola Sundeep

Academic Editor

PLOS ONE